# An Observational Study on the Use of Peripheral Intravenous Lines vs. Central Lines in a Neonatal Intensive Care Unit

**DOI:** 10.3390/children9091413

**Published:** 2022-09-18

**Authors:** Arieh Riskin, Adir Iofe, Donia Zidan, Irit Shoris, Arina Toropine, Rasha Zoabi-Safadi, David Bader, Ayala Gover

**Affiliations:** 1Department of Neonatology, Bnai Zion Medical Center, 47 Golomb Street, P.O. Box 4940, Haifa 31048, Israel; 2Rappaport Faculty of Medicine, Technion—Israel Institute of Technology, Haifa 32000, Israel

**Keywords:** peripheral and central intravenous lines, newborn infants, gestational age, head circumference, weight

## Abstract

Background and Objectives: There is a debate regarding the preferred intravenous (IV) access for newborns. Our aim was to study practices regarding the choice of vascular access and outcomes. Methods: A seven-month prospective observational study on IV lines used in all newborns admitted to Bnai Zion Medical Center’s neonatal intensive care unit (NICU). Results: Of 120 infants followed, 94 required IV lines. Infants born at ≤32 weeks gestation, or with a head circumference ≤29 cm were more likely to require two or more IV lines or a central line for the administration of parenteral nutrition or medications for longer periods. However, central lines (umbilical or peripherally inserted central catheters (PICC)) were not associated with better nutritional status at discharge based on weight z-scores. Only one complication was noted—a central line-associated bloodstream infection in a PICC. Conclusions: Our data suggest preferring central IV access for preterm infants born at ≤32 weeks or with a head circumference ≤29 cm. We encourage other NICUs to study their own data and draw their practice guidelines for preferred IV access (central vs. peripheral) upon admission to the NICU.

## 1. Introduction

Intravenous (IV) therapy is an essential part of the care of a newborn [1]. The need for indwelling IV catheters to provide therapy is considered a standard of care. IV catheters are used to administer continuous parenteral nutrition (PN) and/or medications for preterm and sick full-term infants. The choice of an appropriate vascular access device in newborns depends on the birth weight, the type of infusate, the expected duration of therapy, and the neonatal intensive care unit (NICU)’s policy [2].

Unfortunately, IV access also increases the risk of complications [1]. Repeated attempts to obtain IV access can disrupt the skin’s integrity and cause extravasation [3,4], thus increasing the risk of systemic infections.

In the first few days of life, access to the umbilical vessels is easy and readily available. In order to prevent infections, it is recommended to switch from an umbilical line to another venous access after a certain amount of days (usually after 5–7 days) [5,6]. Usually, the possibilities are either applying a simple peripheral IV (PIV) line or applying a peripherally inserted central catheter (PICC) line [7], each has its advantages and disadvantages [2,8,9]. The insertion of a PIV in newborn infants can be difficult. Appropriate blood vessels to apply intravenous infusion become less available throughout a hospital stay. A PICC is usually inserted when long-term IV medications or PN is necessary after removal of the umbilical vein catheter (UVC), or when UVC insertion has failed. A PICC is inserted in one of the major peripheral veins (i.e., the basilic vein, the cephalic vein, the greater saphenous vein, or less frequently the temporal vein) [10]. PICC lines are aimed to reduce the number of painful IV procedures [7].

A Cochrane systematic review analyzed six trials that recruited 549 infants. One trial showed that use of a PICC was associated with a smaller deficit between prescribed and actual nutrient intake during the trial period (mean difference (MD) −7.1%, 95% confidence interval (CI) −11.02 to −3.2). Infants in the PICC group needed significantly fewer catheters (MD −4.3, 95% CI −5.24, −3.43). A meta-analysis of data from all trials revealed no evidence of an effect on the incidence of invasive infection (typical risk ratio (RR) 0.95, 95% CI 0.72 to 1.25; typical risk difference (RD) −0.01, 95% CI −0.08 to 0.06). The author’s conclusion was that no evidence suggests that PICC use increases the risks of adverse events, particularly invasive infection, although none of the included trials was large enough to rule out an effect on uncommon severe adverse events such as pericardial effusion [11].

Our aim in this study was to describe the practices in our NICU, regarding the different vascular access devices chosen for NICU admissions. Some NICUs have a policy of supporting the use of an early PICC line, while others prefer to use PIVs. We wanted to elucidate the possible outcomes of these different practices, especially nutritional and infectious, and possibly come up with some suggestions to aid the clinical decision on the selection of preferred IV access.

## 2. Methods

This was a prospective observational cohort study. All newborns admitted to the NICU of Bnai Zion Medical Center in Haifa, Israel over a period of 7 months were included (6 November 2017 through 8 June 2018). Bnai Zion hospital NICU is a 20-bed level III NICU with approximately 250 annual admissions.

The hospital’s ethics committee on human research approved the study protocol (Bnai Zion Medical Center ethics committee; Approval Code: 0095-17-BNZ; Approval Date: 7 August 2017). The study complied with the World Medical Association Declaration of Helsinki regarding ethical conduct of research involving human subjects. Being an observational study following a cohort of all infants in the NICU during the study period, the study has been granted an exemption from requiring parents’ written informed consent.

Data were collected daily on: (A) which IV route was being used (UVC, PIV, or PICC) [7,10]; (B) the lifespan of the IV access device used [12]; and (C) the type of fluids, drugs, and blood products that were administered through that line [2]. In our institution, physicians are responsible for the decision whether a neonate requires vascular access. Both physicians and nurses are responsible for peripheral IV insertions. Yet, only physicians or neonatal nurse practitioners insert central vascular access (umbilical or PICC).

Data recorded also included: (A) how many attempts were needed to successfully apply the IV route [13]; (B) the reason to stop using the IV access route (end of treatment, suspected extravasation, etc.); and (C) complications (blood stream infections (BSI) [14], central line-associated BSIs (CLABSI) [13], infection in the area of insertion, thrombosis, etc.). To define blood stream infections, we required two positive blood cultures that were drawn before antibiotic treatment was initiated. If the baby had a central line, it was defined as a catheter-associated infection, even if the two cultures were drawn from peripheral veins, as we do not sample blood from PICCs.

Additionally, we collected nutritional data, including: (A) the time to full enteral feeds (TFEF) of 150 cc/Kg/day [15,16]; and (B) nutritional outcomes as expressed in growth parameters [17,18,19,20] using weight z-scores [21].

Data were collected into an Excel spreadsheet (Microsoft Office, Seattle, WA, USA), and were statistically analyzed using SigmaPlot, version 11.0 (Systat Software Inc., San Jose, CA, USA) and Minitab^®^, version 16.2.2 (Minitab Inc. State College, PA, USA and Coventry, UK). Statistical analysis included descriptive statistics, Mann–Whitney test, and Kruskal–Wallis one-way analysis of variance on ranks (medians) for the comparison of continuous variables between groups with non-parametric distributions, and chi-square test for comparisons of categorical variables. Data were presented as mean ± standard deviation and medians, and *p*-values of less than 0.05 were considered statistically significant. Multiple factors that were significantly different between the study groups and could have affected the outcomes of interest were entered into a multiple linear regression or multiple logistic models to test their effects together. Receiver–operator curves (ROC) were drawn to further study variables of interest and their effects on IV lines, in order to test area under curves (AUC) and to try to define the prediction of cut-off values.

## 3. Results

One hundred and twenty infants were admitted to our NICU during the study period. Their characteristics are presented in Table 1.

First, we analyzed our data by IV lines (Table 2). This analysis includes all instances of venous access, including more than one per patient if needed. For this analysis, if a neonate received an umbilical line, which was later changed to a PICC and/or peripheral venous access, all these events were counted.

Significant factors associated with a need for IV infusion for longer periods in the univariate model, were entered into a multiple linear regression model. In this model, a longer duration of IV was significantly associated with a younger gestational age (R-square 0.19, *p* < 0.001), (or a lower birth-weight (*p* < 0.001), or a smaller head circumference at birth (*p* < 0.001)), as well as with male gender (R-square 0.21, *p* = 0.011) and with more solutions/medications-infused IV (i.e., PN as opposed to only antibiotics) (R-square 0.24, *p* = 0.011). We also ran a multiple logistic regression model to find the variables that were associated with a greater chance that the infant will need a central IV line. The chances that an infant would require a central IV line were significantly increased with a younger gestational age at birth (odds ratio (OR) with an increasing gestational age of 0.89 (95% confidence interval (CI) 0.80; 0.99), *p* = 0.028); with male gender (OR for females 0.21 (95% CI 0.09; 0.46), *p* < 0.001); and with more solutions/medications-infused IV (i.e., PN as opposed to only antibiotics) (OR 2.28 (95% CI 1.43; 3.65), *p* < 0.001).

Then, we analyzed our data by patients (Table 3). In order to run this analysis, we defined a dominant line for each infant, based on the type of intravenous lines (peripheral or central) used for most of his/her hospitalization (i.e., for the longest period). We found that infants on IV took a significantly longer time to reach full enteral feeds (4.3 ± 1.7 days for dominant peripheral and 4.7 ± 2.8 days for a dominant central IV line vs. 3.1 ± 1.4 days in infants without IV lines, *p* = 0.008). However, the difference between infants on PIVs and central lines in the time needed to reach full enteral feeds was not significant.

Table 4 presents the data analysis by patients only for those infants who needed an IV access (94 out of 120).

Significant variables associated with greater chance to require a central line were entered into a multiple logistic regression model. The model showed that the chances that an infant will end up with a dominant central IV line is significantly increased with a smaller head circumference at birth (odds ratio (OR) with a growing head circumference of 0.83 (95% confidence interval (CI) 0.70; 0.98), *p* = 0.028). Then, we drew a receiver–operator curve (ROC) in order to try to define the cut-off value of head circumference at birth of infants who are likely to require a central line (Figure 1A).

In search of other ways to identify those infants who are possible candidates for central IV lines, we did another analysis of our data by patients, but this time we determined for each infant whether he/she required a single IV line or two or more intravenous lines (peripheral or central) (≥2 IV lines). The assumption behind this analysis was that those who required two or more IV lines should be considered as candidates for a central line (Table 5).

Significant variables associated with a greater chance to require two or more IV lines were entered into a multiple logistic regression model. The model showed that the chances that an infant will require two or more IV lines is significantly increased with a younger gestational age at birth (OR with a growing gestational age of 0.86 (95% CI 0.76; 0.98), *p* = 0.023). Then, we drew a ROC curve in order to try to define the cut-off value of the gestational age at birth of infants who are likely to require two or more IV lines during their hospitalization (Figure 1B), and are thus more likely to be considered as candidates for a central IV line.

No other complications beyond the one case of blood stream infection were reported. No infections in the area of IV insertion, thrombosis, or other complications were reported.

Unfortunately, the quality of the data collected for several of the outcome parameters mentioned in the methods were poor and were not suitable for statistical analysis. Most of our IV lines were used for the provision of parenteral nutrition, and thus reaching an adequate amount of enteral nutrition was the reason to stop using IV access. In addition, because of our strict policy regarding the cessation of antibiotic administration after 48 h at the most if blood cultures are negative, and the fact that we did not record any case of an infant with early or late-onset sepsis requiring a prolonged antibiotic course, we can state with a high level of certainty that during the study period we did not have any case of holding IV access just for medications.

## 4. Discussion

Our main finding was that newborn infants born after 32 weeks or less of gestation, or those infants born with a head circumference of 29 cm or less at birth (which is approximately the 50th percentile for a head circumference at 32 weeks gestation), are more likely to require a central IV line for the administration of PN or medications for longer periods. This finding could aid clinicians in the decision of which IV line to insert on the admission of a newborn infant into the NICU. For infants in need for IV access, who are born at 32 weeks gestation or less, or have a head circumference of 29 cm or less (which might be a helpful criterion for intrauterine growth-restricted infants born after more than 32 weeks gestation), a central line should probably be the preferred choice. In the first few days of life, this could probably be a UVC, because the access to the umbilical vessels is easy and readily available and usually requires less skill and experience, yet an initial PICC is an acceptable possibility [22] if trained personnel are available. In order to prevent infections, it is recommended in infants with an initial UVC to switch from an umbilical line to PICC usually after 5–7 days or even less [5,6], if still in need for IV access. Of note is that in our study, head circumference (and not birthweight) was found to be a better intrauterine growth predictor of need for a central IV line, along with the gestational age at birth.

As expected, infants on IV took longer to reach full enteral feeds (Table 3), which was probably associated with the reason they got an IV (for PN) to start with (most commonly prematurity and morbidities associated with it). However, there was no significant difference between infants on PIVs and central lines in the time needed to reach full enteral feeds. This might hint to possible inadequate selection criteria for candidates for central line insertion in our NICU, as the optimal situation would be that infants who are expected to take longer to reach full enteral feeds should have been placed on PN via a central line for longer periods.

Although better weight gain was found in preterm infants with central lines (Table 2, Table 3, and Table 4), this was not associated with significantly better z-scores at discharge from the hospital, which is the ultimate measure for nutritional status. From Table 5, we might speculate on a possible explanation for this observation, because infants in need for two or more IV lines had a worse percentage of change in their weight z-scores, which could have resulted from the fact that many of them who only had PIVs could have benefited from a central IV access with a good intake of PN for a longer period until sufficient enteral intake was secured.

Central IV lines, especially PICCs, lasted for longer periods and were associated with a longer IV duration and longer hospitalization, which was probably associated with the cause for their insertion (Table 2, Table 3 and Table 4). PIVs were associated with more IV lines and thus more trials (Table 2). Unfortunately, PICCs were also associated with more trials, which are related to the PICC insertion technique that is in many ways similar to PIV insertion, requiring more skill and experience compared to UVC insertion. This raises two issues. One is related to the decision to initially insert a UVC if the baby needs an IV access and fulfils the criteria suggested above (i.e., a gestational age of 32 weeks or less or a head circumference of 29 cm or less at birth). UVC provides a readily available central IV access, which requires less skill and experience, and is usually more successful with fewer trials, which are less painful for the infant, to start with. If the infant requires continuation of an IV access beyond the first 3–5 days, a PICC line should be inserted. The second issue is related to the decision of who should be in charge of PICC lines’ insertions, and actually all PIVs. Should it be a designated very experienced skillful IV team [23,24], or should we encourage most physicians, nurse practitioners, and nurses working in the NICU to acquire these skills at the price of less experience and worse outcomes, i.e., more painful trials, as seen in the results from our NICU [25,26]. In the analysis of our results, we could not find differences in the number of attempts of insertion of IV lines between more junior or more experienced medical personnel (physicians or nurses), but our data in this regard were not full, and was thus not presented in the results section or in the tables.

The rate of infections in our study was very low, yet the single episode of BSI was CLABSI in a PICC line, reminding us of the possible consequences of prolonged IV use, or the number of trials associated with PICC line insertion in this study, which was similar to PIVs [13]. However, this should be put in the context of the meta-analysis of data from all trials on PICC line use that revealed no evidence of an effect on the incidence of invasive infections. The meta-analysis authors’ conclusion was that there is no evidence to suggest that PICC use increases the risks of adverse events or invasive infections [11].

The main limitation of our study is that it is based on data from a single center. Practices may vary from one NICU to another depending on the size, the number of preterm infants, the experience of the teams’ members, etc. These could all have effects on IV access choices and success rates. We also did not address the clinical condition and the diagnosis of the infants studied as a factor that could potentially affect the decision to insert an IV line, or the choice of peripheral vs. central IV access. This data could have added more insight on IV use, especially in term or late preterm infants admitted to the NICU, because these infants usually do not need an IV unless they are sick. Because of the relatively small number of infants included in this single center cohort, we could not run an analysis of subgroups, e.g., by gestational age or diagnoses. Another possible limitation was that before this study we did not have strict guidelines in our NICU regarding criteria as to which infant needs IV access and what should be the preferred route—peripheral or central IV. Thus, in many of our infants these decisions were made at the discretion of the attending neonatologist, a factor that could not be evaluated in our statistical analysis. However, it is exactly this problem that led us to investigate this important practice and resulted in easy clinical measures that could guide us as to the preferred IV route.

We believe that our study is important for delivering a message and encouraging other NICUs to study their own data on the use of central vs. peripheral IV lines and outcomes, and based on this to draw conclusions and define their recommendations for IV access best choice and practice.

In conclusion, our data suggest preferring central IV access for preterm infants born at ≤32 weeks or with a head circumference ≤ 29 cm. We encourage other NICUs to study their own data and draw their practice guidelines for preferred IV access (central vs. peripheral) upon admission to the NICU.

## Figures and Tables

**Figure 1 children-09-01413-f001:**
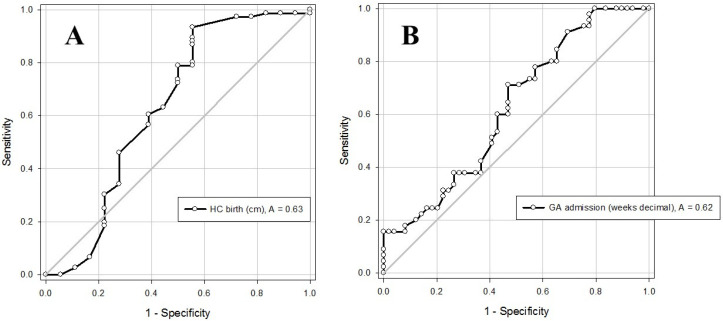
ROC Curves and cut-off values to identify infants who are likely to require central line. (**A**) **By dominant IV line (0 = peripheral, 1 = central):** Area under curve (AUC) of ROC for head circumference (HC) at birth (in centimeters) = 0.63 (95% confidence interval (CI) 0.46–0.81. Using HC cutoff > 29 cm (~the 50% percentile at ~32 weeks’ gestation) the sensitivity is 0.89 (95% CI 0.80–0.95) and the specificity is 0.44 (95% CI 0.21–0.69). (**B**) **By need for two or more IV lines during hospitalization:** AUC of ROC for gestational age (GA) upon admission to the NICU (in weeks) = 0.62 (95% CI 0.51–0.74). Using GA cutoff at birth > 32 weeks the sensitivity is 0.98 (95% CI 0.88–1.00) and the specificity is 0.22 (95% CI 0.12–0.37).

**Table 1 children-09-01413-t001:** Study population demographics (*n* = 120).

Male: Female	57:63 (1:1.1)
Gestational age at birth (weeks)	35.8 ± 3.3 (25.0–41.5) *
Birthweight (gram)	2526 ± 875 (510–5100) *
Head circumference at birth (cm)	32.0 ± 2.8 (21.5–37.0) *
Length of stay in the NICU (days)	18.4 ± 21.7 (2.0–157.0)
Post-menstrual age at discharge (weeks)	38.4 ± 2.7 (35.5–49.9)
Weight at discharge (gram)	2897 ± 728 (1900–5685)
Head circumference at discharge (cm)	33.5 ± 2.0 (31.5–39.1)

Data presented as mean ± SD (range). * 10% (12) were below 32 weeks gestation at birth, 11.7% (14) had birth-weight of 1500 g or lower, and 14.2% (17) had head circumference of 29 cm or less at birth.

**Table 2 children-09-01413-t002:** Analysis of clinical characteristics and outcomes by the type of IV lines used (*n* = 192): This analysis includes all instances of venous access.

	Peripheral Intra-Venous Line (PIV) (*n* = 162)	Umbilical Venous Central Catheter (UVC) (*n* = 19)	Peripherally Inserted Central Catheter (PICC) (*n* = 11)	*p Value* ^‡^
Gestational age at birth (weeks)	35.1 (33.0, 37.0)	36.4 (31.3, 39.5)	31.5 (27.6, 35.3)	*0.086*
Birthweight (grams)	2317 (1850, 3045) *	2250 (1432, 3836)	1500 (1030, 2198) *	* **0.041** *
Head circumference at birth (cm)	32.5 (29.6, 34.0) *	32.5 (27.0, 35.0)	28.5 (25.7, 31.1) *	* **0.013** *
Time to full enteral feeds (days)	4.0 (3.0, 5.0)	5.0 (3.2, 6.7)	4.0 (3.0, 6.2)	*0.356*
Length of stay (hospitalization days)	17.0 (9.0, 26.0) *	25.0 (11.0, 53.0)	35.0 (23.0, 54.0) *	* **0.002** *
Gestational age at discharge (weeks)	37.4 (36.3, 40.3) *	40.4 (38.3, 42.4) *	38.2 (35.2, 41.8)	* **0.048** *
Weight at discharge (grams)	2657 (2395, 3290) *	3310 (2962, 3921) *	2940 (2454, 3766)	* **0.011** *
Head circumference at discharge (cm)	33.0 (32.0, 35.0)	34.5 (32.6, 36.0)	33.5 (32.6, 34.7)	*0.115*
Weight gain (g) during hospitalization (%)	240 (60, 650)	335 (−46, 1575)	1170 (291, 1534)	* **0.027** *
	8.8 (2.3, 30.7)% *	25.0 (−1.3, 104.1)%	72.3 (17.9, 103.4)% *	* **0.010** *
Delta weight z-score at discharge (% change)	−0.45 (−0.69, −0.17)	−0.48 (−0.85, 0.17)	−0.41 (−0.56, 0.34)	*0.523*
	−11.5 (−83.0, 39.1)%	−33.9 (−64.1, −11.2)%	−22.6 (−88.6, −0.1)%	*0.449*
Number of attempts until line insertion	1.0 (1.0, 2.0) †	1.0 (1.0, 1.0) *†	3.0 (2.2, 4.0) *	* **<0.001** *
Line lifespan (hours)	24.0 (12.0, 48.0) *†	72.0 (39.0, 96.0) †	144.0 (96.0, 144) *	* **<0.001** *
Total IV duration (hours)	72.0 (48.0, 96.0) *†	84.0 (65.2, 216.0) †	171 (121.5, 215.5) *	* **<0.001** *
Number of late-onset blood stream infections	0	0	1	* **<0.001** * ^•^

Data are presented as median and interquartile range. ^‡^ Because of non-parametric distribution, normality test failed, and all comparisons were done by Kruskal–Wallis one way analysis of variance on ranks (using medians). If the differences in the median values among the groups were greater than would be expected by chance; there was a statistically significant difference (*p* < 0.05) (marked in bold). To isolate the group or groups that differ from the others, all pairwise multiple comparison procedures were used (Dunn’s method). Pairs that significantly differ from each other are marked by * or †. ^•^ We had only 1 late-onset blood stream infection (BSI) (that was actually central line-associated BSI = CLABSI) on DOL #7 by *Klebsiella Pneumonia* in a central line (PICC) that was inserted on DOL #4. Infant was treated by antibiotics via the central line and infection cleared without having to remove the line. Analysis by chi-square = 16.5 with 2 degrees of freedom (*p* < 0.001).

**Table 3 children-09-01413-t003:** Analysis of clinical characteristics and outcomes by patients’ dominant IV line (*n* = 120).

	Control (No Need for IV Line) (*n* = 26)	Dominant Peripheral IV Lines (*n* = 76)	Dominant Central IV Lines (*n* = 18)	*p Value* ^¥^
Gestational age at birth (weeks)	37.2 ± 2.1 *	35.6 ± 3.0	34.4 ± 5.0 *	* **0.016** *
Birthweight (grams)	2683 ± 627	2508 ± 803	2371 ± 1367	*0.491*
Head circumference at birth (cm)	32.8 ± 1.6 *	32.2 ± 2.5	30.4 ± 4.2 *	* **0.013** *
Time to full enteral feeds (days)	3.1 ± 1.4 *†	4.3 ± 1.7 †	4.7 ± 2.8 *	* **0.008** *
Length of stay (hospitalization) (days)	8.1 ± 4.9 *	17.4 ± 17.9	37.6 ± 35.9 *	* **<0.001** *
Gestational age at discharge (weeks)	38.4 ± 1.9	38.1 ± 2.3	39.8 ± 4.6	*0.068*
Weight at discharge (grams)	2764 ± 733 *	2844 ± 634 †	3311 ± 965 *†	* **0.028** *
Head circumference at discharge (cm)	33.4 ± 1.6	33.4 ± 1.6	33.9 ± 3.5	*0.633*
Weight gain during hospitalization (g) (%)	80 ± 220 *	336 ± 487	940 ± 1089 *	* **<0.001** *
	2.9 ± 7.0% *	23.7 ± 69.4%	84.0 ± 144.8% *	* **0.003** *
Delta weight z-score at discharge (% change)	−0.38 ± 0.49	−0.43 ± 0.59	−0.19 ± 0.61	*0.326*
	−4.5 ± 27.5%	−5.2 ± 69.6%	−1.3 ± 6.4%	*0.969*
Total number of IV lines	0 *†	1.7 ± 0.9 †	2.1 ± 1.0 *	* **<0.001** *
Total IV duration (hours)	0 *†	64.9 ± 41.1 †‡	139.9 ± 71.3 *‡	* **<0.001** *
Number of late-onset blood stream infections	0	0	1	*0.067* •

Data are presented as mean ± standard deviation. ^¥^ All comparisons were done by one way analysis of variance (ANOVA) vs. the control group (without IV lines). If the differences in the mean values among the groups were greater than would be expected by chance, there was a statistically significant difference (*p* < 0.05) (marked in bold). To isolate the group or groups that differ from the others, all pairwise multiple comparison procedures were used (Tukey method) as well as Dunnett’s comparison vs. the control group. Pairs that significantly differ from each other are marked by * or † or ‡. • Chi-square analysis for categorical data.

**Table 4 children-09-01413-t004:** Analysis by of clinical characteristics and outcomes by patients’ dominant IV line, only for those infants who had IV lines (*n* = 94).

	Dominant Peripheral IV Lines (*n* = 76)	Dominant Central IV Lines (*n* = 18)	*p Value* ^‡^
Gestational age at birth (weeks)	35.2	35.7	*0.679*
	(33.4, 38.7)	(31.5, 38.8)	
Birthweight (grams)	2350	1896	*0.275*
	(1950, 3062)	(1435, 3390)	
Head circumference at birth (cm)	32.5	30.5	*0.080*
	(30.5, 34.2)	(28.0, 33.5)	
Time to full enteral feeds (days)	5.0	4.0	*0.829*
	(3.5, 5.0)	(3.0, 7.0)	
Length of stay (hospitalization)(days)	14.5	26.0	* **0.003** *
	(7.0, 24.5)	(12.0, 53.0)	
Gestational age at discharge (weeks)	37.4	40.5	* **0.048** *
	(36.5, 40.2)	(36.9, 42.7)	
Weight at discharge (grams)	2657	3240	* **0.014** *
	(2397, 3267)	(2920, 3850)	
Head circumference at discharge (cm)	33.0	34.0	*0.059*
	(32.0, 34.7)	(33.0, 36.0)	
Weight gain (g) during hospitalization (%)	197 (40, 532)	927 (135, 1500)	* **0.044** *
	7.4 (1.1, 25.7)%	42.8 (4.4, 103.3)%	* **0.033** *
Delta weight z-score at discharge (% change)	−0.48 (−0.71, −0.18)	−0.30 (−0.61, 0.34)	*0.096*
	−1.1 (−59.6, 73.3)%	−20.4 (−56.3, −8.4)%	*0.148*
Total number of attempts for IV lines	3.0	4.5	*0.223*
	(1.0, 5.0)	(2.0, 5.0)	
Total number of IV lines	1.0	2.0	* **0.037** *
	(1.0, 2.0)	(2.0, 3.0)	
Total IV duration (hours)	60.0	123.0	* **<0.001** *
	(36.0, 84.0)	(84.0, 214.0)	
Number of late-onset blood stream infections	0	1	*0.191* •

Data are presented as medians and interquartile ranges. ^‡^ Because of non-parametric distribution, normality test failed, and all comparisons were done by Mann–Whitney test (using medians). If the differences in the mean values among the groups were greater than would be expected by chance, there was a statistically significant difference (*p* < 0.05) (marked in bold). • Chi-square analysis for categorical data.

**Table 5 children-09-01413-t005:** Analysis of clinical characteristics and outcomes by patients, only for those that had IV lines. In this analysis, we compared those who required one IV line vs. those who required two or more IV lines during their NICU course (*n* = 94).

	Single IV Line (*n* = 45)	≥2 IV Lines (*n* = 49)	*p Value* ^‡^
Gestational age at birth (weeks)	35.7	34.5	* **0.039** *
	(34.0, 38.9)	(32.8, 37.3)	
Birthweight (grams)	2567 ± 749	2404 ± 1071	*0.398* *
Head circumference at birth (cm)	32.5	31.5	*0.160*
	(31.0, 34.5)	(29.1, 34.0)	
Time to full enteral feeds (days)	4.0	5.0	*0.236*
	(2.7, 5.0)	(4.0, 5.0)	
Length of stay (hospitalization)(days)	9.0	21.0	* **0.001** *
	(5.0, 22.7)	(11.0, 27.2)	
Gestational age at discharge (weeks)	37.4	38.1	*0.988*
	(36.5, 40.5)	(36.5, 40.7)	
Weight at discharge (grams)	2735	2820	*0.607*
	(2436, 3236)	(2406, 3521)	
Head circumference at discharge (cm)	33.0	33.5	*0.698*
	(32.4, 34.6)	(32.0, 35.1)	
Weight gain during hospitalization (g) (%)	135 (1, 519)	265 (81, 771)	* **0.050** *
	5.6 (0.0, 25.8)%	13.6 (3.2, 43.8)%	* **0.042** *
Delta weight z-score at discharge (% change)	−0.40 (−0.70, −0.14)	−0.47 (−0.70, −0.10)	*0.572*
	18.4 (−17.5, 152.0)%	−33.9 (−94.8, 21.0)%	* **<0.001** *
Total number of attempts for IV lines	1.0	5.0	* **<0.001** *
	(1.0, 2.0)	(3.0, 7.2)	
Total number of IV lines	1.0	2.0	* **<0.001** *
	(1.0, 1.0)	(2.0, 3.0)	
Total IV duration (hours)	48.0	84.0	* **<0.001** *
	(36.0, 72.0)	(60.0, 121.5)	
Number of late-onset blood stream infections	0	1	*1.000* •

Data are presented as medians and interquartile ranges, unless written otherwise. * Analysis by *t*-test. Data are presented as mean ± SD. ^‡^ Because of non-parametric distribution, normality test failed, and all comparisons were done by Mann–Whitney test (using medians), unless written otherwise. If the differences in the mean values among the groups were greater than would be expected by chance, there was a statistically significant difference (*p* < 0.05) (marked in bold). • Chi-square analysis for categorical data.

## Data Availability

The data presented in this study are available on request from the corresponding author.

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
