# Peer review of "An Observational Study on the Use of Peripheral Intravenous Lines vs. Central Lines in a Neonatal Intensive Care Unit"

_children, 2022, doi:10.3390/children9091413_

Round 1

Reviewer 1 Report

In their prospective observational study, Riskin and colleagues studied the use and success of intravenous vascular access in their neonatal intensive care unit. They aimed at detecting factors to identify neonates at need for central venous access, which is in important topic for neonatal practice. The limitations of the study, which are accurately acknowledged, are the presented single-centre experience, the rather low number of included patients and that the patients’ clinical condition and disease were not considered in the analysis.

Although the manuscript is well structured and written, there are some major aspects that require further explanation and/or modification:

1)  Methods: To be able to interpret the findings, the reader needs more information about your NICU: What level of care does it provide? How many neonates receive treatment every year? How many beds does it contain? Who decides whether a neonate requires vascular access and who is responsible to establish it?

2) Unfortunately, the presentation of the results is rather confusing. According to the abstract, 94/120 neonates required vascular access. Conversely, Table 1 reports peripheral venous access in 162 cases, umbilical venous access in 19 and peripherally inserted central catheters (PICC) in 11 neonates. Do these numbers include all instances of venous access (i.e. more than one per patient, if needed)? Please explain this by adding an explanatory paragraph at the beginning of the results. Furthermore, please explain the following in the methods section: If a neonates received an umbilical line, which was later changed to a PICC and/or peripheral venous access, how did you count these events?

3) Several outcome parameters that were named in the methods are not reported in the results:

- “the type of fluids, drugs and blood products that were administered” This is important, because the required medication may impact the required vascular access.

- “the reason to stop using the IV access route”

- “Complications” The authors report about blood streams infections, but not about “infection in the area of insertion” and “thrombosis”, as mentioned in the methods.

4) The authors report as their major finding that preterm neonates below 32 weeks of gestation are more likely to require central venous access, which is not surprising and something that I would totally agree with from a clinical standpoint. However, the included neonates obviously were rather mature, with a median gestational age of 31.5 weeks in the PICC group according to Table 1. How many neonates below 32 weeks of gestation were actually included in this study? For this purpose, I recommend adding a table with demographic information to the results. And I am also wondering whether this study did actually include enough very preterm neonates for this statistical analysis/conclusion?

5) Please add a conclusion at the end of the discussion.

Minor aspects are as follow:

Abstract – Background and Objectives: The sentence beginning with “There is a debate …” is confusing. As umbilical venous access is generally considered to be central and the analysis mainly differentiates between peripheral and central vascular access, I suggest deleting the part after “… for newborns”.

Introduction, page 2, paragraph 3: Please provide a reference for the recommended practice to switch from umbilical venous catheter to “another venous access” after 5-7 days.

Introduction, page 2, paragraph 3: Please use the abbreviation PICC consistently after its first introduction.

Introduction, page 2, paragraph 4: I suggest adding “A” or “One” before “Cochrane systematic review”.

Introduction, page 3, paragraph 5: The authors state that they try to “possibly come up with guidelines”, yet such an attempt would be challenging based on a single-centre observational study and it is not presented in the manuscript. I suggest removing this study aim.

Methods, page 3, paragraph 2: Please add the number of the respective ethics committee approval.

Methods, page 4, paragraph 4: How were blood stream infections and catheter-associated infections defined?

Methods, page 4, paragraph 5: There are different names for the Mann-Whitney test: Wilcoxon-Mann-Whitney test, Mann-Whitney-U test or Wilcoxon rank sum test – please use either one.

Methods, page 4, paragraph 5: Please change “Data was …” to “Data were …”.

Results, Tables 1, 3 & 4: Please change “number of trials” to “number of attempts”.

Results, Table 1: Please change “line survival length” to “line lifespan” or appropriate.

Results, page 6, paragraph 1: Please change “The chances that an infant will …” to “The chances that an infant would …”.

Discussion, page 11, paragraph 1: Please explain why you argue that central venous access in the “first few days of life” should be provided by an umbilical line (and not by an initial PICC).

Discussion, page 14: Please change “… a factor that could not have been …” to “… a factor that could not be …”.

References: Please label reference 7 as a thesis.

Author Response

We wish to thank the the reviewer for his.her thoughtful comments.

We have extensively revised our manuscript according to his.her comments as listed below and as marked in the attached manuscript text by the "track changes" function.

We hope you will find the revised manuscript suitable for publication.

Sincerely,

Arieh Riskin MD, MHA, PhD on behalf of all authors.

Point by point address of the reviewer's 1 comments and suggestions:

  1. The data regarding our NICU was added to paragraph 1 in the methods section (page 4), and the question regarding the decision and the insertion of vascular access is addressed in paragraph 3 of the methods section (page 5).
  2. All the questions raised in comment 2 were clarified in the second paragraph of the results section before new table 2 (former table 1) (page 7).
  3. We have addressed as best as we could the important issues related to the other outcome measures as requested by the reviewer. These are detailed at the end of the results section on pages 14.
  4. We accepted the very good and thoughtful comment by the reviewer and added new table 1 with the demographics of our study population. As mentioned at the bottom of this table we had 12 infants (10%) who were below 32 weeks gestation at birth (page 6).
  5. We added the conclusion at the end of the discussion (page 17-18).

Minor comments:

  1. Abstract – The sentence was changed as suggested by the reviewer (page 2).
  2. Introduction – We added two references for this recommended practice (references 5, 6 on page 3).
  3. Introduction – The text was corrected to the abbreviation PICC (page 3).
  4. Introduction – The text was corrected as suggested "… A Cochrane systematic review" (page 3).
  5. Introduction – Indeed, guidelines could not be the appropriate outcome for a single center study. We changed it to "possibly come up with some suggestions …" which seems more appropriate in this context (page 4).
  6. Methods: The details on the approval code etc. were sent to the editorial board as requested, and were now added into the text as requested by the reviewer (page 4).
  7. Methods: To define blood stream infections we require two positive blood cultures that were drawn before antibiotic treatment was initiated. If the baby has a central line it is defined as catheter-associated infection, even if the two cultures are drawn from peripheral veins, as we do not sample blood from PICCs (page 5).
  8. Methods: The name Mann-Whitney test is now used along the text (page 5 and in the comments to the tables).
  9. Methods: The text was corrected to "Data were presented …" (page 6).
  10. Results: The text in tables (2, 4 & 5, previously 1, 3 & 4) was corrected to "number of attempts".
  11. Results: The text in table 2 (previously table 1) was corrected to "line lifespan" as suggested (page 7).
  12. Results: The text was corrected to "The chances that an infant would require" (page 8).
  13. Discussion: The text was corrected to reflect different practices, and the reason why we usually use umbilical veins: "In the first few days of life, this could probably be a UVC, because the access to the umbilical vessels is easy and readily available and usually requires less skill and experience, yet an initial PICC is an acceptable possibility [22] if trained personnel is available (page 15).
  14. Discussion: The text was corrected to "a factor that could not be …" (page 17).
  15. References: I have changed the definition of reference 9 (previously 7) to thesis, and the new format it appears in the text is according to MDPI style downloaded to EndNote from the journal website. However, the word thesis does not appear in the reference. A managing editorial consultation is welcomed.

Reviewer 2 Report

the authors should have a more descriptive title on all tables. For example, table 1 : analysis by IV lines. Analysis of what? The same happens all over the manuscript.

The legend of the figure 1 is completely out of place. Legends should be under the figures. Please check the journal guidelines. Moreover, the legends are not descriptive enough and the axis of the graphs doesn't say much. Sensitivity and specificity? It's really hard to understand the figures just by themselves.

References 9, 10 and 22 are in the wrong format. Please check the journal's guidelines.

Author Response

We wish to thank the the reviewer for his.her thoughtful comments.

We have extensively revised our manuscript according to his.her comments as listed below and as marked in the attached text by the "track changes" function.

We hope you will find the revised manuscript suitable for publication.

Sincerely,

Arieh Riskin MD, MHA, PhD on behalf of all authors.

Following is a point by point responses to reviewer's 2 comments and suggestions:

  1. We have revised all the titles of our tables to make them more descriptive and explanatory. We hope this is adequate.
  2. As for figure 1, we moved the legend to be under the figure as requested. However, we reviewed the legend title and subtitles A & B, and did not see how we could further improve them. As for the legend of the axis, they are usually marked these way in ROC curves, and the explanation as to what sensitivity and specificity they address usually appear in the legends as we did. However, we welcome the reviewer or editors' suggestions regarding the text of the legends of the title, subtitle or axis.
  3. References: These references (11, 12, 25 previously 9, 10, 22) were corrected. All references are formatted automatically to MDPI style downloaded to EndNote from the journal website. Originally, they were all found in PubMed and collected into one EndNote library. If you think they are still not appropriate, we would welcome a managing editorial consultation and help in this matter.

Round 2

Reviewer 1 Report

I would like to thank the authors for their quick and comprehensive response. All my questions were answered adequately and the implemented changes helped further improving the manuscript. From my point of view, there are only two minor aspects left that should be addressed:

Methods, page 5, paragraph 2: I suggest changing “Physicians accept the decisions regarding whether a neonate requires a vascular access” to “In our institution, physicians are responsible for the decision whether a neonate requires vascular access”.

Discussion, page 17, paragraph 4: Please change “In conclusion, our data suggests …” to “In conclusion, our data suggest …”.

Author Response

We are happy that the reviewer found our answers to his.her questions adequate. We wish to thank the reviewer again for the time and effort he.she put in the review of our manuscript. Indeed, the implemented changes as the reviewer suggested helped  improve the manuscript significantly.

We corrected the two minor aspects left on pages 5 and 17 as the reviewer suggested (please find the revised version attached).

Sincerely,

Arieh Riskin on behalf of the authors

Reviewer 2 Report

I believe that the current version of the manuscript can be published in this journal.

Author Response

We wish to thank the reviewer again for his.her comments.

Attached is the final second revision of the manuscript.

Sincerely,

Arieh Riskin on behalf of the authors
